# Two-Dimensional Selenium Nanosheet-Based Sponges with Superior Hydrophobicity and Excellent Photothermal Performance

**DOI:** 10.3390/nano12213756

**Published:** 2022-10-26

**Authors:** Hongyan Chen, Mengke Wang, Weichun Huang

**Affiliations:** 1Engineering Training Center, Nantong University, Nantong 226019, China; 2School of Chemistry and Chemical Engineering, Nantong University, Nantong 226019, China

**Keywords:** Xene, selenium, sponges, hydrophobic, photothermal

## Abstract

Photothermally assisted superhydrophobic materials play an important role in a variety of applications, such as oil purification, waste oil collection, and solar desalination, due to their facile fabrication, low-cost, flexibility, and tunable thermal conversion. However, the current widely used superhydrophobic sponges with photothermal properties are usually impaired by a high loading content of photothermal agents (e.g., gold or silver nanoparticles, carbon nanotubes), low photothermal efficiency, and require harmful processes for modification. Here, a one-pot, simple composite consisting of two-dimensional (2D) selenium (Se) nanosheets (NSs) and commercially used melamine sponge (MS) is rationally designed and successfully fabricated by a facile dip-coating method via physical adsorption between 2D Se NSs and MS. The loading content of 2D Se NSs on the skeleton of the MS can be well controlled by dipping cycle. The results demonstrate that after the modification of 2D Se NSs on the MS, the wettability transition from hydrophilicity to hydrophobicity can be easily achieved, even at a very low loading of 2D Se NSs, and the highly stable photothermal conversion of the as-fabricated composites can be realized with a maximum temperature of 111 ± 3.2 °C due to the excellent photothermal effect of 2D Se NSs. It is anticipated that this composite will afford new design strategies for multifunctional porous structures for versatile applications, such as high-performance solar desalination and photothermal sterilization.

## 1. Introduction

The Group VI element selenium (Se), one of Xenes (phosphorus [1,2], tellurene [3,4], bismuthene [5,6], antimonene [7,8], etc.), is an important semiconductor that offers intriguing properties, including anisotropic thermal conductivity, excellent photoconductivity, and superior piezoelectric and thermoelectric response [9,10]. Se has been reported to have excellent photothermal efficiency, which has been widely applied in biomedical applications, such as photothermal radiotherapy [11] and imaging-guided synergistic chemo-photothermal therapy [12,13]. The large photothermal effect, cost-effective fabrication, and relatively low cytotoxicity of Se nanostructures make them competitive candidates in many applications, such as waste oil collection, oil purification, solar desalination, and photothermal-assisted antibacterial application.

The rapid development of the 5G era and the multifunctionality of miniaturized equipment have yielded many impressive benefits for humans [5,14,15,16]. Among multifunctional devices, photothermally assisted superhydrophobic materials play an important role in the fields of waste oil collection, oil purification, and solar desalination due to their facile fabrication, low-cost, flexibility, and tunable thermal conversion [17,18,19]. For example, in 2021, Han et al. [18] reported high-efficiency photothermal conversion material MXene nanosheets (NSs) and low thermal conductivity silica (SiO_2_) coated on a hydrophilic poly(tetrafluoroethylene) (HPTFE) membrane by a commercial continuous spraying system to fabricate a SiO_2_/MXene/HPTFE Janus membrane, and demonstrated that the film had high stability, light absorption, salt resistance, and self-cleaning ability. Moreover, in 2020, Li et al. [20] passivated 2D black phosphorus (BP) NSs with hydrophobic SiO_2_ by hydrolytic co-condensation of 3-aminopropyl-triethoxysilane and tetraethoxysilane, which exhibited high efficiency and stability in solar evaporation without sacrificing the intrinsic properties of BP NSs. In the past decade, many researchers have focused on the superhydrophobic materials with high thermal efficiency and broad-spectrum light absorption to maximum the light-to-heat conversion [21,22,23,24]. As a typical kind of 3D materials, melamine sponge (MS) was widely used as a universal substrate in the fields of oil/water separation and solar desalination due to its large adsorption capacity, high stability, and low cost [25,26]. However, the intrinsic hydrophilicity and extremely low light-to-heat transition efficiency greatly restrict the work efficiency of MS in related industries [27,28]. In this scenario, hydrophobization by low surface energy materials and doping of photothermal agents are common strategies for the improved performance of the MS [29].

Although photothermally assisted hydrophobic materials have undergone great progress, the reported fabrication of the multifunctional materials is usually time-consuming, complicated, environmentally unfriendly, and difficult to be realized for industrial production. For example, the reported modifiers for the hydrophobization of MXene NSs and BP NSs, such as polydimethylsiloxane and 1H,1H,2H,2H-perfluorooctyltriethoxysilane, are usually harmful to the environment and organisms, which goes against the principles of green chemistry. Therefore, the development of a one-pot and simple fabrication of superhydrophobic materials with both excellent hydrophobicity and high thermal efficiency in an environmentally friendly manner is crucial.

In this study, two-dimensional (2D) Se nanosheets (NSs) were successfully fabricated by a facile LPE method, and then directly employed for the production of composites, abbreviated as Se@MS, composed of the 2D Se NSs and melamine sponge (MS) via physical absorption by a dip-coating method. The loading content of 2D Se NSs on the skeleton of the MS is easily controlled by the dipping cycle. Three kinds of the Se@MSs are obtained with the loading content of the 2D Se NSs of 2.8 ± 0.5 wt%, 4.4 ± 0.5 wt%, and 5.1 ± 0.8 wt%. After the modification of the 2D Se NSs, the wettability of the pristine MS rapidly changes from hydrophilicity to hydrophobicity, even at a low loading of the 2D Se NSs (2.8 ± 0.5 wt%). The photothermal result demonstrates that the as-fabricated Se@MS has an excellent photothermal efficiency with a maximum photothermal temperature of 111 ± 3.2 °C, significantly higher than that of pristine MS (55 ± 2.1 °C) under the same conditions. In addition, at a high loading of 2D Se NSs (5.1 ± 0.8 wt%), the photothermal conversion declines mainly due to the relatively serious aggregation of the 2D Se NSs on the skeleton of the MS, which largely reduces the efficient specific surface area. Due to the facile and environmentally friendly fabrication of Se@MS with low cost, rapid wettability transition from hydrophilicity to hydrophobicity, and excellent photothermal efficiency, it is anticipated that this Se@MS can pave the way for new designs of multifunctional porous structures for versatile applications, such as high-performance solar desalination and photothermal sterilization.

## 2. Experimental Section

### 2.1. Materials

Se powder (99.9%) and isopropyl alcohol (IPA, 99.9%) were purchased from Shanghai Macklin Biochemical Co., Ltd. (1288 Canggong Rd, Shanghai Chemical Industry Park, Shanghai 201424, China) and used as received without further purification. The commonly used MS was purchased from the local market in Nantong, Jiangsu, China. Double-distilled deionized water was used in the process of water contact angle characterization.

### 2.2. Fabrication of 2D Se NSs

A facile LPE method was used to fabricated 2D Se NSs, as previously reported [30]. In brief, bulk Se powder was first dispersed into IPA solvent with a concentration of 8 mg mL^−1^, and then the bulk Se/IPA slurry was exfoliated in a sonication bath with a built-in water-cooling system for 48 h. The sonication temperature was fixed at 5 °C and sonication power was controlled at around 300 W. Afterwards, the slurry was centrifuged at a speed of 3000 rpm for 10 min, and the supernatant containing 2D Se NSs was slowly decanted to another centrifugation tube and further collected at a centrifugation speed of 18,000 rpm for 20 min. The centrifugation precipitate was dried at 80 °C under vacuum overnight.

### 2.3. Fabrication of Se@MS

Se@MS was fabricated by a simple dip-coating method. In brief, 10.8 mg commonly used MS was cut into a cube (dimensions: 2 cm × 2 cm × 2 cm) and completely cleaned by sonication in DI water and IPA for 5 min in sequence. Afterwards, the dried MS was immersed into a 1 mg mL^−1^ Se NS/IPA dispersion for 10 min. After the MS was removed from the dispersion, the reductant Se NSs on the MS was completely removed by squeezing, and the as-obtained Se NS-absorbed MS was dried in a vacuum oven at 80 °C for 2 h; the obtained sample is abbreviated as Se@MS-1 (2.8 ± 0.5 wt%). Repeat the procedure to obtain the high loading of Se NSs on the MS, and the as-obtained Se NS-absorbed MS is abbreviated as Se@MS-N, where N denotes the cycle of soaking process. Note that each process was repeated at least three times to report an average value. In this work, Se@MS represents the as-obtained Se@MS-1, Se@MS-2, and Se@MS-3, unless otherwise specified. The weight percentage of Se NSs on the MS was determined through the following equation: weight percentage (%) = (*w*_1_ − *w*_0_)/*w*_0_ × 100%, where *w*_1_ represents the sample weight after the modification of 2D Se NSs and *w*_0_ represents the sample weight of the pristine MS.

### 2.4. Characterization

An Ultima IV X-ray diffractometer was used to collect X-ray diffraction (XRD) signals of bulk Se and 2D Se NSs. Transmission electron microscopy (TEM, FEI Tecnai G2 F30) and high-resolution TEM (HRTEM) with an acceleration voltage of 200 kV were employed to determine the morphology and atomic arrangement of the as-fabricated 2D Se NSs, respectively. Energy dispersive spectroscopy (EDS) was performed using a FEI Tecnai G2 F30 TEM equipped with the Oxford EDAX EDS system. The morphology of the as-fabricated Se@MS was characterized by scanning electron microscopy (SEM, ZEISS Gemini SEM 300) at an acceleration voltage of 5.0 kV, and the element mapping analysis and EDS of the as-fabricated Se@MS were also characterized using an EDAX system attached on the SEM at an acceleration voltage of 15.0 kV. Size distribution of the Se NSs was evaluated by the dynamic light scattering (DLS). The N_2_ adsorption–desorption curves and the pore size distribution were measured by a Brunauer–Emmett–Teller (BET) specific surface area and porosity analyzer (TriStar Ⅱ 3020, Quantachrome Autosorb IQ MP). An optical CA instrument (JGW-360B) was employed to characterize the water contact angle (CA) using 5 μL droplets at room temperature, and the reported value denotes an average of three measurements. The photothermal performance was studied under illumination from a xenon short arc lamp (Appendix A, wavelength: 800–1100 nm) at a distance of 20 cm, and the infrared thermal photographs were taken by a thermal imaging camera (ST 9450A+, Smart Sensor).

## 3. Results and Discussion

Figure 1 presents the structural characterization of the 2D Se NSs as fabricated by a facile LPE method. The XRD pattern (Figure 1a) shows that the as-fabricated nanostructures have the characteristic peaks of bulk Se, in good agreement with the standard HA (JCPDS No. 86-2246) diffraction peaks, and no impurity peaks (such as SeO_2_) were observed. The TEM image (Figure 1b) shows that the as-fabricated Se nanostructures present 2D structural morphology with a lateral size of 180–380 nm. Size distribution of the Se NSs was evaluated by the dynamic light scattering (DLS). Appendix A shows an average diameter of Se NSs measured by DLS is 480 ± 2.5 nm, which is in good accordance with the TEM results. The HRTEM image (Figure 1b inset) shows a lattice fringe of 0.30 nm, which can be indexed to the (101) plane of the Se crystal [31]. The selected area electron diffraction (SAED) pattern (Figure 1c) also confirms the successful fabrication of the 2D Se NSs by a LPE method.

The fabrication of the 2D Se NS-based MS was performed by a simple dip-coating method via physical adsorption. As the number of dipping cycles increased, the loading of the 2D Se NSs on the MS gradually increased (Figure 2a), i.e., the weight percentages of the 2D Se NSs on the MS are 2.8 ± 0.5 wt%, 4.4 ± 0.5 wt% and 5.1 ± 0.8 wt%, for Se@MS-1, Se@MS-2, and Se@MS-3, respectively. The weight percentage of the selected Se@MS-2 was also confirmed by EDS (N: 41.8 ± 0.6 wt%, C: 38.5 ± 0.5 wt%, O: 15.1 ± 0.4 wt%, Se: 4.0 ± 0.1 wt%, Figure 2b). The difference in weight percentage was due to the different characterization techniques. The change in morphology of Se MS on the skeleton with the increasing weight percentage of Se can be seen in Figure 3a–f. The 2D Se NSs are well distributed on the surface of MS skeleton, and with an increase in dipping cycle, the density of the 2D Se NSs on the MS remarkably increases. It is noted that the aggregation of 2D Se NSs in Figure 3e,f can be mainly attributed to the high loading of the 2D Se NSs on the MS (5.1 ± 0.8 wt%) for the Se@MS-3 sample. Due to the high porosity of the Se@MS, the surface areas are measured to be 55.45 ± 0.9 m^2^ g^−1^; 44.69 ± 1.2 m^2^ g^−1^; 40.05 ± 0.8 m^2^ g^−1^, and 37.40 ± 1.1 m^2^ g^−1^ for the pristine MS, Se@MS-1, Se@MS-2, and Se@MS-3, respectively (Appendix A), in good agreement with the SEM result. In addition, EDS analysis (Figure 3g) of the Se@MS-2 illustrates that the elemental C, N, O, and Se are uniformly distributed, and the location of the elemental Se is in good accordance with the loading position in the SEM image.

The wettability of a water droplet on the surface of the pristine MS and Se@MS was studied, as shown in Figure 4. It can be seen in Figure 4a,b and Appendix A that the water droplet can quickly spread and penetrated into the pristine MS when it was dropped onto the surface of the sample. On the contrary, for the Se@MS samples, the water droplet clearly beads up, demonstrating the strong in-air hydrophobicity after the modification of 2D Se NSs. Besides, it can be also observed that when the pristine MS and as-fabricated Se@MS-2 were immersed into a glass of water, the pristine MS quickly sinks into the bottom while the as-fabricated Se@MS-2 keep floating on the surface due to its excellent hydrophobicity (Appendix A). The optical images of the MS and Se@MS confirm that the wettability of the commonly used MS can be distinctly switched by surface modification of the 2D Se NSs at a relatively low loading (Figure 4a). The wetting transition of MS from hydrophilicity to hydrophobicity after Se NSs treatment is induced by the hydrogen bonding effect between the exposed Se atoms on the Se NSs and the N-H groups in the skeleton of MS. The formation of -N-H∙∙∙Se changed the surface chemical states of MS and greatly lowered its surface energy, and thus the wetting property of MS was reversed [32]. The similar wettability reversion of MS was also reported in the formation of metal-ion-induced cross-linkage [27,33] and -N-H∙∙∙F hydrogen bonds [17]. Notably, for the three studied Se@MS samples, the water CA increases with the dipping cycle, i.e., the water CAs of the Se@MS-1, the Se@MS-2, and the Se@MS-3 are 137 ± 0.8°, 151 ± 0.3°, and 144 ± 0.7°, respectively (Figure 4b). In addition, the excellent stability of water contact angle for the as-fabricated Se@MS-2 (Figure 4c) over one month indicates that the Se@MS holds great promise for practical applications. The rapid wettability transition from hydrophilicity to hydrophobicity by the modification of the 2D Se NSs on the MS is expected to have great potential in anti-fouling and self-cleaning smart equipment in the biomedical field.

Given the superior photothermal effect of Xenes, such as phosphorene [1,34,35], bismuthene [6,36], and MXenes [37,38,39], the photothermal conversion of the as-fabricated Se@MS with excellent hydrophobicity was also studied, as shown in Figure 5. It can be observed in Figure 5a–e that at a fixed light power density, the temperatures of Se@MS-2 and Se@MS-3 remarkably increase under illumination for 300 s compared with that of the pristine MS, e.g., under light illumination with a power density of 0.8 W cm^−2^ for 300 s, the temperatures of the Se@MS-2 and Se@MS-3 quickly increase to 102 ± 4.8 °C and 86 ± 3.7 °C, respectively, significantly higher than that (57 ± 2.3 °C) of the pristine MS (Figure 5d), verifying that the Se NSs employed in the Se@MS indeed have an excellent photothermal efficiency. Note that the Se@MS-2 achieved the best photothermal effect compared with Se@MS-1 and Se@MS-3, which can be mainly ascribed to the suitable loading of Se NSs and uniform distribution on the surface of MS skeleton. Here, the loading of 2D Se NS in the Se@MS-1 is very low (2.8 ± 0.5 wt%), leading to an inconspicuous temperature change in comparison with the pristine MS at the studied power density range (0.2–1.0 W cm^−2^), while the loading of 2D Se NS in the Se@MS-3 is fair yet the relatively serious aggregation of 2D Se NSs make a remarkable reduction in the efficient photothermal conversion due to the apparent reduction in specific surface area. Moreover, it can be observed that the temperature of the Se@MS distinctly increases with the power density, i.e., the temperatures of the Se@MS-3 are 57 ± 1.4 °C (0.2 W cm^−2^, Figure 5a), 61 ± 3.9 °C (0.4 W cm^−2^, Figure 5b), 80 ± 2.1 °C (0.6 W cm^−2^, Figure 5c), 86 ± 3.7 °C (0.8 W cm^−2^, Figure 5d), and 93 ± 2.8 °C (0.8 W cm^−2^, Figure 5e), respectively, which demonstrates that the photothermal effect can be rationally controlled by the external power density and loading of 2D Se NSs. In addition, the temperature cycle of the Se@MS-2 at a high light power density of 1.0 W cm^−2^ (Figure 5f) shows that the as-fabricated Se@MS has a very stable photothermal conversion, even at a high photothermal temperature of 111 ± 3.2 °C. The high thermal conversion efficiency of Se NSs can be attributed to the narrow band gap energy (*E*_g_) due to nonradiative relaxation. The 2D Se NSs capture the solar light, and stimulate the fall of electrons back to the low-energy states and thus the energy is released through radiating photons or nonradiative phonons. In the nonradiative mode, the heat is produced when the phonon interacts with the lattice, establishing a temperature gradient based on the optical absorption and electron–hole recombination feature [11,12].

The infrared thermal photos of the as-fabricated Se@MS in one on/off temperature cycle can be seen in Figure 6. Upon illumination at 1.0 W cm^−2^, the surface temperature of the as-fabricated Se@MS-2 rapidly rises up from room temperature (29.8 ± 1.6 °C) to a maximum (111 ± 3.2 °C) within 300 s, and quickly recovers when the light illumination is off, suggesting the rapid photoresponse behavior and excellent reproducibility under strong light illumination. Table 1 briefly summarizes the photothermal performance of representative reported photothermal agents, such as multiwalled carbon nanotubes (MWCNTs) [19,40] and BP NSs [20]. Notably, the hydrophobic Se@MS in this work is superior/comparable to these photothermal agents for photothermal efficiency. Apart from that, it is noted that there is no obvious morphology change for the as-fabricated Se@MS before and after photothermal investigation at such a higher light power density (Appendix A). This, combined with excellent hydrophobicity of the Se@MS, indicates that the sample Se@MS-2 is an ideal biomedical candidate for both high-performance self-cleaning and photothermal conversion, which has great potential for practical biomaterial scaffold and biological packaging materials.

## 4. Conclusions

In this study, 2D Se NSs with a lateral size of 180–380 nm were successfully fabricated by a facile LPE method, and were directly employed to fabricate Se@MS by a simple dip-coating method via physical absorption between 2D Se NSs and MS. Through the absorption of 2D Se NSs by the MS in sequence, Se@MSs with different loadings of 2D Se NSs are obtained, as evidenced by SEM result that the loadings of 2D Se NSs on the MS are 2.8 ± 0.5 wt%, 4.4 ± 0.5 wt% and 5.1 ± 0.8 wt%, respectively, and the elemental Se is uniformly distributed on the MS skeleton. The apparent wettability transition from hydrophilicity (0°) to hydrophobicity (137 ± 0.8°~151 ± 0.3°) has been achieved after the modification of 2D Se NSs on the MS, even at an extreme loading, indicating that the Se@MSs have great potential in self-cleaning applications. In addition, the high photothermal conversion of all Se@MSs is obtained due to the high photothermal effect of the Se NSs on the MS skeleton, but the Se@MS-3 exhibits relatively poor photothermal performance due to the severe aggregation at a high loading of 2D Se NSs. The highest temperature can reach up to 111 ± 3.2 °C at 1.0 W cm^−2^ with good stability. Because of the facile fabrication of 2D Se NSs and Se@MS, rapid wettability transition from hydrophilicity to hydrophobicity, quick photothermal response, and high stability, it is anticipated that this Se@MS holds great promises in various applications, including solar desalination, photothermal sterilization, etc.

## Figures and Tables

**Figure 1 nanomaterials-12-03756-f001:**
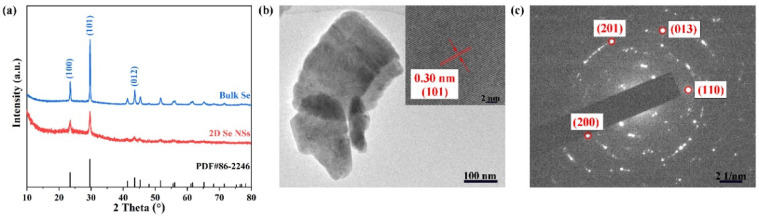
The structural characterization of the as-fabricated 2D Se NSs. (**a**) XRD patterns of bulk Se and 2D Se NSs; (**b**) TEM image of 2D Se NSs, inset shows the HRTEM image; and (**c**) SAED pattern.

**Figure 2 nanomaterials-12-03756-f002:**
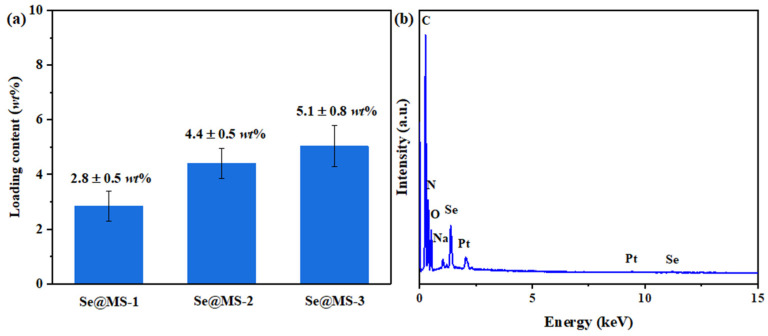
The analysis of Se content in the as-fabricated Se@MS. (**a**) Weight percentage analysis of Se content in the Se@MS-1, Se@MS-2, and Se@MS-3, and (**b**) element content of the Se@MS-2 measured by EDS.

**Figure 3 nanomaterials-12-03756-f003:**
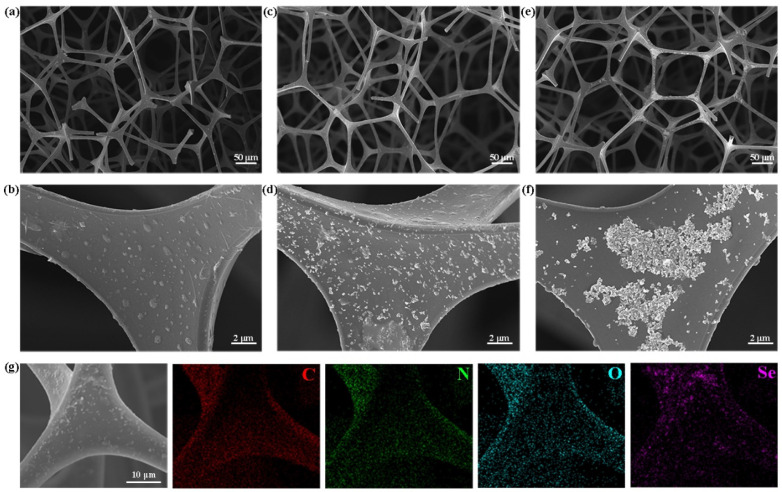
SEM image of the as-fabricated Se@MS. (**a**) Se@MS-1 and (**b**) its enlarged view; (**c**) Se@MS-2 and (**d**) its enlarged view; (**e**) Se@MS-3 and (**f**) its enlarged view; and (**g**) SEM image of the Se@MS-2 and the elemental mapping images.

**Figure 4 nanomaterials-12-03756-f004:**
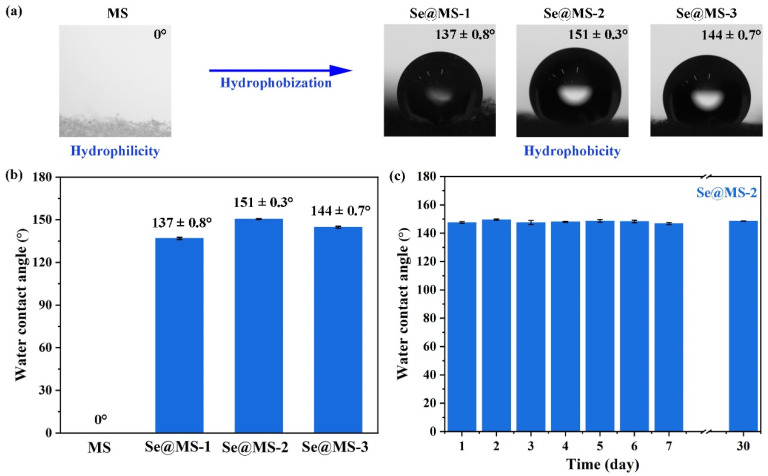
Hydrophobic performance of pristine MS and Se@MS. (**a**) Optical images of the pristine MS, Se@MS-1, Se@MS-2, and Se@MS-3; (**b**) water contact angle of the pristine MS, Se@MS-1, Se@MS-2, and Se@MS-3; and (**c**) stability test of the Se@MS-2 measured by water contact angle over one month.

**Figure 5 nanomaterials-12-03756-f005:**
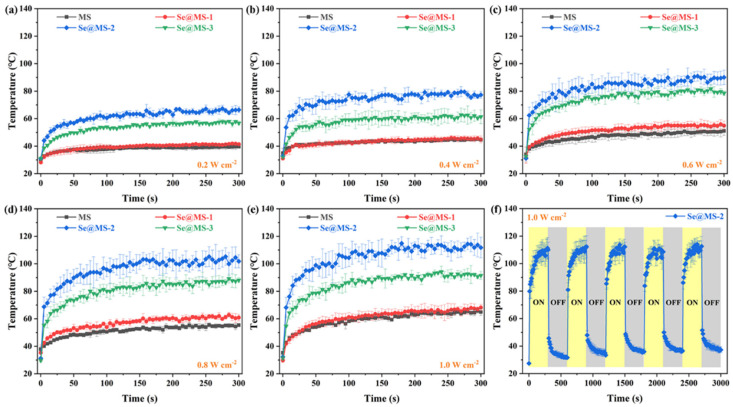
Photothermal performance of Se@MS. Surface temperature variations of pristine MS, Se@MS-1, Se@MS-2, and Se@MS-3 under illumination at different power densities: (**a**) 0.2 W cm^−2^; (**b**) 0.4 W cm^−2^; (**c**) 0.6 W cm^−2^; (**d**) 0.8 W cm^−2^; and (**e**) 1.0 W cm^−2^. (**f**) Five on/off temperature cycles of Se@MS-2 under illumination at 1.0 W cm^−2^.

**Figure 6 nanomaterials-12-03756-f006:**
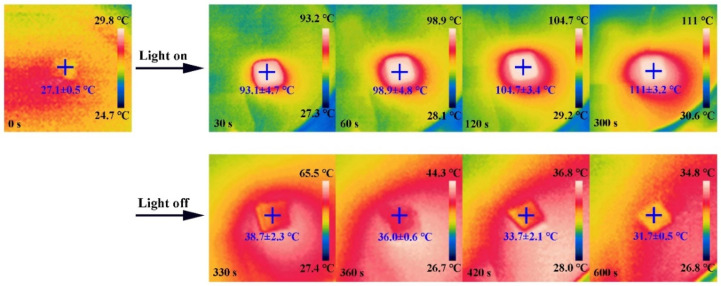
Infrared thermal pictures of Se@MS-2 under illumination at 1.0 W cm^−2^ for one temperature on/off cycle.

**Table 1 nanomaterials-12-03756-t001:** Photothermal performance comparison of the recently reported materials.

Photothermal Agents	Substrate	Loading Content	Light Source	Power Density (W cm^−2^)	Temperature (°C)	Temperature Rising Time (s)	Ref.
MWCNTs	PVP	77 wt%	Simulator solar	0.1	41	900	[19]
MWCNTs	PU	20.6 wt%	Simulator solar	0.1	88	300	[40]
Copper nanorods	-	Aqueous dispersion	980 nm laser	2.5	40	2000	[41]
MXene Ti_3_C_2_T*_x_*	Poly(tetrafluoroethylene) (PVDF)	0.47 mg cm^−2^	Xe lamp	0.1	44.7	3600	[18]
MXene Ti_3_C_2_T*_x_*	MS	0.1 wt%/2.0 wt%	Simulator solar	1.0	47/53	300	[17]
BP NSs	Silica	~60 wt%	Simulator solar	0.1	32.8	1500	[20]
CuS nanospheres	Polyacrylamide, and carboxymethyl cellulose	12 mg mL^−1^	Simulator solar	0.1	47.2	1800	[42]
Se NSs	MS	2.8 wt%	Simulator solar	0.2	57	300	This work

## Data Availability

Not applicable.

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
