# Peer review of "Two-Dimensional Selenium Nanosheet-Based Sponges with Superior Hydrophobicity and Excellent Photothermal Performance"

_nanomaterials, 2022, doi:10.3390/nano12213756_

Round 1
Reviewer 1 Report (New Reviewer)
This work systematically investigates the photothermal and wettability properties of Se nanosheet-decorated sponge. Some results are very useful but not complete. There are still some questions needed to be addressed (see comments). Thus, I would recommend a major revision of this manuscript.
1. As the author mentioned in the abstract “the widely used superhydrophobic sponges with photothermal performance usually suffer from harmful modification process…” The Se is widely known as a highly carcinogenic element. How to avoid it affecting human health during use?
2. The red words in Figure 1b and 1c are not clear, please change it.
3.The most serious issues are Figure 5, Figure S6 and Table 1. As the data offered in the manuscript, the sponge with Se nanosheet is white, and it means that the white object can reflect all light on its surface, including IR light, while IR camera only record IR light and then temperature is shown. How did the author confirm that the IR camera recorded the real temperature but not the IR light combined with thermal emission and IR light reflection. Because the light source is also IR light, the recorded temperature should be much higher than the real temperature. I think that why the reported temperature is higher than others’ works at the same condition. I would like to recommend the authors to measure the optical property and thermal emissivity to make sure the IR camera can record the accurate temperature.
Author Response
Reviewers’ comments and our responses:
This work systematically investigates the photothermal and wettability properties of Se nanosheet-decorated sponge. Some results are very useful but not complete. There are still some questions needed to be addressed (see comments). Thus, I would recommend a major revision of this manuscript.
Response: We thank the reviewer for his/her positive feedback on the overall quality of the manuscript.
Q1: As the author mentioned in the abstract “the widely used superhydrophobic sponges with photothermal performance usually suffer from harmful modification process…” The Se is widely known as a highly carcinogenic element. How to avoid it affecting human health during use?
Response: Thank you very much for pointing this out! The World Health Organization recommends a daily selenium dose of 30 mg to 40 mg for adults [Biomed. Pharmacother. 2003, 57, 134–144; J. Geochem. Explor. 2010, 107, 206–216; Food Chem. 2011, 124, 1050–1055], and also emphasizes that a selenium dose of 400 mg/day is harmless [Nutrition 2013, 29, 713–718]. Detectable signs of selenium toxicity were observed in patients who received 3200 mg/day. Therefore, the as-fabricated 2D Se NS-based sponges have negligible effect on the human health during use due to the very low loading content of the 2D Se NS (2.8-5.1 wt%) on the sponges.
Q2: The red words in Figure 1b and 1c are not clear, please change it.
Response: Thank you very much for pointing this out! The red words in Figure 1b and 1c have been changes as following:
Q3: The most serious issues are Figure 5, Figure S6 and Table 1. As the data offered in the manuscript, the sponge with Se nanosheet is white, and it means that the white object can reflect all light on its surface, including IR light, while IR camera only record IR light and then temperature is shown. How did the author confirm that the IR camera recorded the real temperature but not the IR light combined with thermal emission and IR light reflection. Because the light source is also IR light, the recorded temperature should be much higher than the real temperature. I think that why the reported temperature is higher than others’ works at the same condition. I would like to recommend the authors to measure the optical property and thermal emissivity to make sure the IR camera can record the accurate temperature.
Response: Thank you very much for your nice suggestion! According to the review’s suggestion, we have referred to related literatures about the surface temperature on white objects (Nat. Nanotechnol. 2021, 16, 1281-1291; J. Hazard. Mater. 2021, 403, 124090; ACS Appl. Mater. Interfaces 2021, 13, 47302-47312; ACS Appl. Mater. Interfaces 2021, 13, 21175-21185; Biomaterials 2021, 276, 121007; ACS Appl. Nano Mater. 2021, 4, 5230-5239; etc.). As shown in the above-mentioned references, the photothermal performance of all the white samples was evaluated through the IR thermal camera. Based on the same evaluation method with others’ works, the higher surface temperature in our work is attributed to the excellent photothermal effect of Se nanosheets instead of the usage of IR camera.

Reviewer 2 Report (New Reviewer)
Review of the Manuscript Nanomaterials-1960949
1. English text should be improved and Abstract, Introduction, Results and Conclusions should be rewritten.
2. If possible, please add results from Raman spectroscopy, XPS, pore volumes and pore size distribution.
3. Please add DOI of the references.
Author Response
Reviewers’ comments and our responses:
Q1: English text should be improved and Abstract, Introduction, Results and Conclusions should be rewritten.
Response: Thank you very much for pointing this out! The presentation has been carefully polished, and the Abstract, Introduction, Results and Conclusion have been re-organized and rewritten with a more formal format.
Q2: If possible, please add results from Raman spectroscopy, XPS, pore volumes and pore size distribution.
Response: Thank you very much for your nice suggestion! The related results listed below are in good agreement with the previously published (Adv. Funct. Mater. 2020, 30, 2003301).
Figure R1. Raman spectroscopy of Se NSs.
Figure R2. XPS spectroscopy of Se 3d.
Figure R3. BJH adsorption pore volume distribution curves.
Q3: Please add DOI of the references.
Response: Thank you very much for pointing this out! The each DOI of the reference was added.

Reviewer 3 Report (New Reviewer)
1) The title needs to be rephrased as in the manuscript 2D Se NS term is used. Though MS is a 3D foam, the title suggests that Se NS is 3D.
2) In the abstract section, line 14; please correct the term "physic". Similarly, dipping coating should be "dip coating". Please correct.
3) In the intro section, lines 26-27; please write something relevant to the concerned field and not a generalized technology thing.
4) Overall, the introduction section needs to be modified. The second paragraph..."Group VI element selenium (Se)....." should be at the beginning. More discussion on MS, its hydrophobic properties, and related applications is required.
5) Fabrication process of Se@MS is not properly explained. How did you confirm the wt% of 2D Se NSs?
6) What is the acceleration voltage of TEM/HRTEM?
7) DLS is not an ideal method to evaluate the average size of the 2D sheets. This should have been calculated from the TEM images themselves. Overall, the TEM analysis is very poor as anything conclusive cannot be reached.
8) Figure 2a seems irrelevant as the number of dipping cycles is not mentioned.
9) A scientific discussion on the wetting properties of MS and Se@MS is completely missing. How did the loading of 2D Se NSs affect the wetting behavior of MS? I believe that both MS and Se are intrinsically hydrophilic, then it is strange to see the superhydrophobic behavior of Se@MS samples without using any low surface energy coating.
10) Similarly, a scientific discussion on the photothermal efficiency of Se@MS composites is missing. The authors have only explained the obtained results. Fig.5 caption; powder should be power.
Author Response
Reviewers’ comments and our responses:
Q1: The title needs to be rephrased as in the manuscript 2D Se NS term is used. Though MS is a 3D foam, the title suggests that Se NS is 3D.
Response: Thank the review for pointing this out! The Title was revised as “2D Selenium Nanosheet-Based Sponges with Superior Hydrophobicity and Excellent Photothermal Performance”.
Q2: In the abstract section, line 14; please correct the term "physic". Similarly, dipping coating should be "dip coating". Please correct.
Response: Thank the review for pointing this out! The corresponding terms have been corrected.
Q3: In the intro section, lines 26-27; please write something relevant to the concerned field and not a generalized technology thing.
Response: Thank the review for your nice suggestion! The corresponding description “The highly photothermal effect, cost-effective fabrication and relatively low cytotoxicity of Se nanostructures endow them competitive candidates in many applications, such as waste oil collection, oil purification, solar desalination, and photothermal-assisted antibacterial application” on page 4 line 17.
Q4: Overall, the introduction section needs to be modified. The second paragraph..."Group VI element selenium (Se)....." should be at the beginning. More discussion on MS, its hydrophobic properties, and related applications is required.
Response: Thank the review very much for your nice suggestion! Considering the overall layout of this manuscript, we persist that the description “Group VI element selenium (Se) as a number of Xenes (phosphorus [13,14], tellurene [15,16], bismuthene [3,17], antimonene [18,19], etc.) is an important semiconductor that offers intriguing properties, including anisotropic thermal conductivity, excellent photoconductivity, superior piezoelectric and thermoelectric response [20,21]” stay in its original position, and we have reasonably modified this paragraph for better quality.
Q5: Fabrication process of Se@MS is not properly explained. How did you confirm the wt% of 2D Se NSs?
Response: Thank the review for pointing this out! The weight percentage of Se NSs was determined through the following equation by weighting the sample before and after the loading of Se NSs:
|
Weight percentage (%)=(w1-w0)/w0×100% |
Where w1 represents the sample weight after Se NSs modification, w0 denotes the sample weight before Se NSs modification. The corresponding description has been added on page 6 line 22.
Q6: What is the acceleration voltage of TEM/HRTEM?
Response: Thank the review for pointing this out! The acceleration voltage of TEM/HRTEM is 200 kV.
Q7: DLS is not an ideal method to evaluate the average size of the 2D sheets. This should have been calculated from the TEM images themselves. Overall, the TEM analysis is very poor as anything conclusive cannot be reached.
Response: Thank the reviewer for pointing this out! According to the reviewer’s suggestions, more TEM images of Se NSs were provided as follows. From the TEM images, the Se NSs have a lateral size range of 200-500 nm.
Q8: Figure 2a seems irrelevant as the number of dipping cycles is not mentioned.
Response: Thank the review for pointing this out! The dipping cycles of Se@MS-1, Se@MS-2, and Se@MS-3 were 1, 2, and 3 cycles, respectively. The relevant description was clarified in the Experimental section as “…, and the as-obtained Se NS-absorbed MS is abbreviated as Se@MS-N, where N denotes the cycle of soaking process.”
Q9: A scientific discussion on the wetting properties of MS and Se@MS is completely missing. How did the loading of 2D Se NSs affect the wetting behavior of MS? I believe that both MS and Se are intrinsically hydrophilic, then it is strange to see the superhydrophobic behavior of Se@MS samples without using any low surface energy coating.
Response: Thank the review for pointing this out! The wetting transition of MS from hydrophilicity to hydrophobicity after Se NSs treatment could be due to the hydrogen bonding formation between the exposed -NH- group on the MS and Se atoms on the surface of 2D Se NSs. The formation of -N-H∙∙∙Se changed the surface chemical states of MS and greatly lower its surface energy, and thus the wetting property of MS was reversed. The detailed discussion of the mechanism can also be found in the following references: CCS Chem. 2020, 2 191-202; J. Phys. Chem. A 2019, 123. 5995-6002.
Q10: Similarly, a scientific discussion on the photothermal efficiency of Se@MS composites is missing. The authors have only explained the obtained results. Fig.5 caption; powder should be power.
Response: Thank you for your nice suggestion! Considering the different light-matter interaction mechanisms in electromagnetic radiation, two kinds of mechanisms based on 2D nanomaterials can be divided to contribute efficient photothermal conversion, including the metallic materials with localized plasmonic heating and semiconductors with nonradiative relaxation. In this work, high thermal conversion efficiency of Se NSs can be attributed to the narrow band gap energy (Eg) due to the nonradiative relaxation. The 2D Se NSs capture the solar light, and stimulate electrons fall back to the low energy states and thus the energy is released through radiating photons, or nonradiative phonons. In the nonradiative mode, the heat is produced when the phonon interacts with the lattice, establishing a temperature gradient based on the optical absorption and electron-hole recombination feature. The corresponding references have been cited.
- Zheng, N.; Wang, Q.; Li, C.; Wang, X.; Liu, X.; Wang, X.; Deng, G.; Wang, J.; Zhao, L.; Lu, J. Responsive Degradable Theranostic Agents Enable Controlled Selenium Delivery to Enhance Photothermal Radiotherapy and Reduce Side Effects. Adv. Healthcare Mater. 2021, 10, 2002024.
- Liu, X.; Wang, Y.; Yu, Q.; Deng, G.; Wang. Q.; Ma, X.; Wang, Q.; Lu, J. Selenium Nanocomposites as Multifunctional Nanoplatform for Imaging Guiding Synergistic Chemo-Photothermal Therapy. Colloid. Surf. B 2018, 166, 161-169.
In addition, the caption of Fig. 5 was corrected.

Round 2
Reviewer 1 Report (New Reviewer)
I have checked the author’s reply again. The reply to Q1 and Q2 is fine. But for my most concerned question 3, the authors did not answer my question directly, and they can’t offer the optical and thermal emission properties of the samples. In addition, none of the literatures offered by the authors can support their findings, and most of them measured black material.
Especially, they offered their previous work as a proof, where ~53°C is achieved under 1 W cm-2 for 300 s illumination. This result is reasonable, while in this work, they can obtain 111 °C with the same condition. These two results are contradictory to each other. But I still suggest them use a thermal couple to check the real temperature if they can’t measure optical and thermal emission properties of the samples. Therefore, I would not recommend it publish in this journal.
Author Response
Dear Editor,
Thank you for handling our manuscript (ID: nanomaterials-1960949), and we also would like to thank our reviewer for their valuable comments. We have revised the manuscript strictly and accordingly. We hope that you and the reviewer are satisfied with our revision, and are happy to answer any further questions.
Reviewers’ comments and our responses:
Q1: I have checked the author’s reply again. The reply to Q1 and Q2 is fine. But for my most concerned question 3, the authors did not answer my question directly, and they can’t offer the optical and thermal emission properties of the samples. In addition, none of the literatures offered by the authors can support their findings, and most of them measured black material.
Especially, they offered their previous work as a proof, where ~53°C is achieved under 1 W cm-2 for 300 s illumination. This result is reasonable, while in this work, they can obtain 111 °C with the same condition. These two results are contradictory to each other. But I still suggest them use a thermal couple to check the real temperature if they can’t measure optical and thermal emission properties of the samples. Therefore, I would not recommend it publish in this journal.
Response: Thank you very muck for pointing this out! The queries of the reviewer are answered from the following three aspects:
- The thermal emission rate (ε) of MS is 0.95 which is provided by the supplier. It is noteworthy that roughness is a key parameter in the determination of the thermal emission rate of materials. For example, the thermal emission rate of highly polished iron plate is only 0.28, whereas the thermal emission rate of frosted iron plate is as high as 0.94. The high thermal emission rate of MS is induced by its large surface roughness and high porosity, which can absorb most of incident light rather than reflection.
- Considering the reviewer’s suggestions, the surface temperature of Se@MS prepared in this paper and the surface temperature of our previously reported MXene@MS were remeasured under the same conditions and the results were provided in Figure R1. The thermal emission rate of the IR camera was set at a same value of 0.95. The remeasured temperature also coincides with our previous results. Notably, the surface temperatures of 53°C and 111 °C are not contradictory as the reviewer claimed, and the photothermal performances of Se NSs and MXenes are different.
Figure R1. IR thermal images of (a) Se@MS and (b) MXene@MS are measured under the same conditions.
- As the reviewer’s suggestions, the temperature of Se@MS was also checked using a thermocouple thermometer and the results were provided in Figure R2. The surface temperature of Se@MS at 1.0 W cm-2 measured by a thermocouple thermometer can reach ~105℃, which is only slightly lower than the temperature measured by an IR camera, possibly due to the different characterization techniques.
Figure R2. Surface temperature variations of Se@MS measured by a thermocouple thermometer at 1.0 W cm-2.
Thank you again for your considerations!
Sincerely,
Yours,
E-mail: huangweichun@ntu.edu.cn
Professor of School of Chemistry and Chemical Engineering, Nantong University, Nantong 226019, Jiangsu, China
Oct. 19th, 2022

Reviewer 3 Report (New Reviewer)
The quality of the manuscript has not been improved significantly. The revisions are unsatisfactory as the introduction is still not very informative and the discussion should have been more detailed.
Author Response
Dear Editor,
Thank you for handling our manuscript (ID: nanomaterials-1960949), and we also would like to thank our reviewer for their valuable comments. We have revised the manuscript strictly and accordingly. We hope that you and the reviewer are satisfied with our revision, and are happy to answer any further questions.
Reviewers’ comments and our responses:
Q1: The quality of the manuscript has not been improved significantly. The revisions are unsatisfactory as the introduction is still not very informative and the discussion should have been more detailed.
Response: Thank you very much for pointing this out! The introduction section was revised as the reviewer’s suggestions, “Group VI element selenium (Se) as a number of Xenes (phosphorus [1,2], tellurene [3,4], bismuthene [5,6], antimonene [7,8], etc.) is an important semiconductor that offers intriguing properties, including anisotropic thermal conductivity, excellent photoconductivity, superior piezoelectric and thermoelectric response [9,10]. Se has been reported with excellent photothermal efficiency, which was widely applied in the biomedical applications, such as photothermal radiotherapy [11], imaging guiding synergistic chemo-photothermal therapy [12,13]. The highly photothermal effect, cost-effective fabrication and relatively low cytotoxicity of Se nanostructures endow them competitive candidates in many applications, such as waste oil collection, oil purification, solar desalination, and photothermal-assisted antibacterial application” on page 3 line 2; and “In the past decade, many researches focused on the superhydrophobic materials with highly thermal efficiency and broad-spectrum light absorption to maximum the light-to-heat conversion [21-24]. As a typical kind of 3D materials, melamine sponge (MS) was widely used as a universal substrate in the fields of oil/water separation and solar desalination due to its large adsorption capacity, high stability, and low cost [25,26]. However, the intrinsic hydrophilicity and extremely low light-to-heat transition efficiency greatly restrict the work efficiency of MS in related industries [27,28]. In this scenario, hydrophobization by low surface energy materials and doping of photothermal agents are common strategies for the improvement of performance of MS [29]” on page 4 line 1; and “Although photothermally assisted hydrophobic materials have achieved great progress, yet the reported fabrication of the multifunctional materials is usually time-consuming, complicated and environmentally unfriendly, difficult to be realized for industrial production. For example, the reported modifiers for the hydrophobization of MXene NSs and BP NSs, such as polydimethylsiloxane and 1H,1H,2H,2H-perfluorooctyltriethoxysilane, are usually harmful to the environment and organisms, which goes against green chemistry. Therefore, a one-pot and simple fabrication of superhydrophobic materials for both excellent hydrophobicity and highly thermal efficiency in an environmentally friendly manner is very crucial to be developed” on page 4 line 11.
In addition, the detailed discussion of the wettability and the photothermal performance parts was also added in the manuscript, “The wetting transition of MS from hydrophilicity to hydrophobicity after Se NSs treatment is induced by the hydrogen bonding effect between the exposed Se atoms on the Se NSs and the N-H groups in the skeleton of MS. The formation of -N-H∙∙∙Se changed the surface chemical states of MS and greatly lower its surface energy, and thus the wetting property of MS was reversed [32]. The similar wettability reversion of MS was also reported in the formation of metal-ion-induced cross-linkage [27,33] and -N-H∙∙∙F hydrogen bonds [17]” on page 10 line 12, and “The high thermal conversion efficiency of Se NSs can be attributed to the narrow band gap energy (Eg) due to the nonradiative relaxation. The 2D Se NSs capture the solar light, and stimulate electrons fall back to the low energy states and thus the energy is released through radiating photons, or nonradiative phonons. In the nonradiative mode, the heat is produced when the phonon interacts with the lattice, establishing a temperature gradient based on the optical absorption and electron-hole recombination feature [11,12]” on page 13 line 2.
We thank the reviewer again for nice suggestions!
Thank you again for your considerations!
Sincerely,
Yours,
E-mail: huangweichun@ntu.edu.cn
Professor of School of Chemistry and Chemical Engineering, Nantong University, Nantong 226019, Jiangsu, China
Oct. 19th, 2022

Round 3
Reviewer 1 Report (New Reviewer)
OK, now it is fine.
Author Response
Thank you for your nice suggestion.
Reviewer 3 Report (New Reviewer)
The revised manuscript is now in better shape to be accepted for publication.
Author Response
Thank you very much for your recommendation.
This manuscript is a resubmission of an earlier submission. The following is a list of the peer review reports and author responses from that submission.
Round 1
Reviewer 1 Report
The manuscript under consideration deals with an interesting method to prepare Se nanosheets for functionalizing commercial sponges with high hydrophobic and photothermal properties. The paper is well written dividing the worth of experimentals in proper sections and it is supported by a sufficient number of references. Despite some of the properties are well defined (photothermal), the wettability part shows a lack of fundamental approach to properly define the expected features. In facts these systems have to be regarded as heterogeneous surfaces at high porosity, so parameters like roughness and surface energy have to be studied to support such behaviour. Again static contact angle is poorly meaningful for these systems. Few other points also reveal an unaccurate paper like the lack of size distribution of the particles,, a potential self-cleaning activity that should be checked over long time.
Reviewer 2 Report
The manuscript, entitled "3D Selenium Nanosheet-Based Sponges with Superior Hydrophobicity and Excellent Photothermal Performance," is very nice. The authors report very interesting experimental results. They precisely measured the change in hydrophilicity of melamine sponges modified with two-dimensional Se nanosheets. The materials appear to have been manufactured very carefully. The devices fabricated by them also showed very good properties in high thermal conversion under light irradiation. These properties rationalize the application of this material as a biomaterial. Therefore, this paper will be of interest to surface scientists as well as biomaterials researchers. I recommend that this manuscript be accepted for publication in Nanomaterials.
1) If possible, I would like to see a reference to the mechanisms the authors have found for the properties of this material from an atomistic point of view. For example, the following papers are theoretical studies on the wettability of 2D materials. It might be helpful to refer to these and comment on the importance of hydrogen bonding and dispersion forces at the interface.
ACS Omega 2019, 4, 4491-4504; Langmuir 2021, 37, 11351-11364.
2) What is the origin of the high thermal conversion efficiency of Se nanosheets? Is it vibrational relaxation of photo-excited states? It would be helpful to the reader if the authors could add any references.
3) Why does the title of this paper contain "3D" instead of "2D"?